# Reporting Quality of Studies Developing and Validating Melanoma Prediction Models: An Assessment Based on the TRIPOD Statement

**DOI:** 10.3390/healthcare10020238

**Published:** 2022-01-26

**Authors:** Isabelle Kaiser, Katharina Diehl, Markus V. Heppt, Sonja Mathes, Annette B. Pfahlberg, Theresa Steeb, Wolfgang Uter, Olaf Gefeller

**Affiliations:** 1Department of Medical Informatics, Biometry and Epidemiology, Friedrich Alexander University of Erlangen-Nuremberg, 91054 Erlangen, Germany; isabelle.kaiser@fau.de (I.K.); katharina.diehl@fau.de (K.D.); annette.pfahlberg@fau.de (A.B.P.); 2Mannheim Institute of Public Health, Social and Preventive Medicine, Medical Faculty Mannheim, Heidelberg University, 68167 Mannheim, Germany; 3Department of Dermatology, University Hospital Erlangen, 91054 Erlangen, Germany; markus.heppt@uk-erlangen.de (M.V.H.); theresa.steeb@uk-erlangen.de (T.S.); 4Department of Dermatology and Allergy Biederstein, Faculty of Medicine, Technical University of Munich, 80802 Munich, Germany; sonja.mathes@mri.tum.de

**Keywords:** reporting quality, TRIPOD, prediction models, melanoma

## Abstract

Transparent and accurate reporting is essential to evaluate the validity and applicability of risk prediction models. Our aim was to evaluate the reporting quality of studies developing and validating risk prediction models for melanoma according to the TRIPOD (Transparent Reporting of a multivariate prediction model for Individual Prognosis Or Diagnosis) checklist. We included studies that were identified by a recent systematic review and updated the literature search to ensure that our TRIPOD rating included all relevant studies. Six reviewers assessed compliance with all 37 TRIPOD components for each study using the published “TRIPOD Adherence Assessment Form”. We further examined a potential temporal effect of the reporting quality. Altogether 42 studies were assessed including 35 studies reporting the development of a prediction model and seven studies reporting both development and validation. The median adherence to TRIPOD was 57% (range 29% to 78%). Study components that were least likely to be fully reported were related to model specification, title and abstract. Although the reporting quality has slightly increased over the past 35 years, there is still much room for improvement. Adherence to reporting guidelines such as TRIPOD in the publication of study results must be adopted as a matter of course to achieve a sufficient level of reporting quality necessary to foster the use of the prediction models in applications.

## 1. Introduction

Guidelines are a ubiquitous tool in all areas of healthcare to provide evidence-based guidance on how to act appropriately according to current knowledge. The cornerstone of healthcare guidelines, which allows for the synthesis of empirical evidence from the scientific literature on a specific topic, is the proper publication of studies that address the topic in question. To appropriately evaluate a study in the process of research synthesis and for inclusion in the body of scientific evidence that influences the contents of the guideline, two prerequisites exist: First, the publication of a scientific study has to contain all relevant aspects of the design, conduct, and analysis in the necessary detail. Second, the results have to be presented comprehensively. Therefore, a specific subtype of guidelines for researchers has emerged, namely reporting guidelines, which aim to improve the quality of reporting of scientific studies in the medical literature by providing advice on what items should be included in a publication. In 1994, the first version of the CONSORT (CONsolidated Standards Of Reporting Trials) statement, a reporting guideline for randomized parallel group clinical trials, was published [1] and triggered a long-lasting development in which more and more reporting guidelines were created for different types and areas of medical research, such as STROBE (STrengthening the Reporting of OBservational studies in Epidemiology) for observational studies in epidemiology [2], PRISMA (Preferred Reporting Items for Systematic Reviews and Meta-Analyses) for systematic reviews and meta-analyses [3], STARD (STAndards for Reporting of Diagnostic accuracy) for diagnostic accuracy studies [4], SPIRIT (Standard Protocol Items: Recommendations for Interventional Trials) for the definition of standard protocols for clinical trials [5] and COREQ (COnsolidated criteria for REporting Qualitative research) for qualitative research [6], to name just a few of the best-known reporting guidelines.

In 2015, the TRIPOD (Transparent Reporting of a multivariate prediction model for Individual Prognosis Or Diagnosis) statement was published, a reporting guideline that addresses publication of studies describing diagnostic and prognostic prediction models [7]. TRIPOD consists of 22 items that are considered essential for informative reporting of studies developing and/or validating multivariable prediction models. Transparent reporting of these items enables the assessment of the generalizability and risk of bias, as well as the replication of the published models by other researchers. In order to enhance objectivity and to ensure consistent measurement of reporting quality a TRIPOD adherence assessment form was developed in 2019 [8]. It provides guidance for extracting the relevant information and calculating summary scores to determine the degree of adherence to TRIPOD.

One clinical domain, where a high number of studies reporting prediction models have been published during the last 35 years, is the field of cutaneous melanoma. As melanoma is an aggressive malignancy that tends to metastasize beyond its primary site, detection in its early stages is essential for the successful treatment of the disease [9]. The importance of early diagnosis of melanoma combined with its rising incidence over the last decades [10] has fueled the demand for melanoma risk prediction. Our objective was to investigate the reporting quality and compliance with TRIPOD in this special segment of prediction studies.

## 2. Materials and Methods

### 2.1. Study Selection and Eligibility Criteria

In our TRIPOD rating we included all studies developing and validating models for predicting the individual risk of occurrence of cutaneous melanoma. As the basis for the set of studies to be assessed, we used a recent systematic review on melanoma prediction modeling [11] updating two earlier systematic reviews on the topic [12,13]. One eligibility criterion for that systematic review had been that all included studies provide either absolute risk or risk scores, or report mutually adjusted relative risks for primary cutaneous melanoma. Furthermore, the studies had to use a multivariable prediction model and a well-defined statistical method for the development of their models. More details on the search strategy are described in [11].

As the end of the search period for the systematic review was 31 January 2020, we updated the literature search to ensure that our TRIPOD rating included all relevant studies. Specifically, the forward snowballing technique was performed on all three systematic reviews on the topic [11,12,13] for the time interval since the most recent systematic review, that is, February 2020 to August 2021. Forward snowballing is an efficient search strategy that explores citations to specific reference papers and thus looks forward in time when performing a search among citations [14]. Furthermore, an electronic literature search in PubMed using the same search string as in [11] was conducted for the same time interval.

### 2.2. TRIPOD Rating

The reporting quality of each study was assessed by six independent reviewers (I.K., K.D., M.V.H., S.M., T.S., O.G.) using the “TRIPOD Adherence Assessment Form” (www.tripod-statement.org/ (accessed on 21 December 2021)) for a uniform rating and scoring. The reviewer panel was multidisciplinary and consisted of reviewers with methodological, clinical and public health backgrounds and different levels of experience. For data collection, an web-based input tool was created using the software SoSci Survey [15]. All six reviewers rated all 42 studies. Disagreements between the reviewers regarding the assessment of the individual items were resolved in ten consensus meetings. Furthermore, two independent referees (A.B.P and W.U.) decided in case of sustained disagreements.

In total, the TRIPOD checklist contains 22 items related to different parts of the publication: title and abstract (items 1 and 2), introduction (item 3), methods (items 4 through 12), results (items 13 through 17), discussion (items 18 through 20) and other information (items 21 and 22). Ten of the 22 items are split into subitems, resulting in a total of 37 TRIPOD components, see Table 1. Those 12 items without subitems and the 25 subitems contain one or more elements which are mostly scored as either “yes” or “no”. For some elements, there is also the response option “referenced” if the requested information is contained in another publication and the authors provide the reference to this publication. Another response option for specific elements is “not applicable” if the element deemed to be not applicable to a specific situation.

### 2.3. Calculation of TRIPOD Adherence Scores

We used the published scoring algorithms as provided in [8] to quantify the adherence to TRIPOD. If all elements of a particular TRIPOD component are adequately addressed, meaning they are answered with either “yes” or “referenced”, adherence to this TRIPOD component is scored as “1”. Otherwise, non-adherence is scored as “0”. An overall TRIPOD adherence score was calculated as the sum of TRIPOD components divided by the total number of applicable TRIPOD components for the corresponding study report. Item 21—providing information about the availability of Appendix A for the publication—is not taken into account for the score calculation [8]. Furthermore, the number of applicable components depends on the study type, as some of the components do not apply to all study types. The TRIPOD statement covers three types of studies: (i) those that solely report model development, (ii) those that combine development and external validation of a prediction model, and (iii) those that describe solely external validation of an already published model [7]. For studies solely describing model development the sub-/items 10c, 10e, 12, 13c, 17 and 19a do not apply. Therefore, the maximum number of applicable TRIPOD components for the score calculation is 29 for development studies. Subitems that are not rated for validation studies are 10a, 10b, 14a, 14b, 15a and 15b, which again results in a total of 30 applicable TRIPOD components. For studies describing both development and external validation, all 36 sub-/items apply. Additionally, five sub-/items (5c, 10a, 11, 14b, 17) can be rated as “not applicable” by the reviewers reducing the denominator for the TRIPOD adherence score in these cases.

The overall adherence per TRIPOD sub-/item is defined as the number of studies that adhered to a specific item divided by the number of studies in which the item is applicable.

### 2.4. Statistical Analysis

All adherence scores are reported as percentages. Descriptive statistical analyses were performed to describe the score distribution per study and per sub-/item. We further examined whether a temporal effect regarding the reporting quality of risk prediction studies exists. Therefore, we used beta regression, as the adherence scores are restricted to the interval (0,1) [16]. The beta regression is modeled using mean and precision parameters. In a first step, we determined the possible relationship between completeness of reporting as dependent variable and year of publication as only independent variable of the model. In a second step, we added journal subject category (categorical) and journal impact factor (continuous) as independent variables in the mean parametrization of the multivariable regression model. To demonstrate the impact of the subject categories, we calculated the model adjusted mean TRIPOD overall adherence scores for the mean pattern of other variables in the model. Journal subject category and impact factor of 2020 were extracted from the 2021 Journal Citation Reports^®^ [17]. For journals with multiple categories we selected the subject category listed first (e.g., for the journal “Melanoma Research”, which has been assigned to the categories “Oncology”, “Dermatology” and “Medicine, Research & Experimental”, we used the category “Oncology”).

Group comparisons, e.g., between studies reporting solely model development and studies reporting both model development and external validation, were done by the exact version of the non-parametric Mann–Whitney-U test with a significance level of 0.05. All statistical analyses were performed using the R software [18]. Beta regression modeling was implemented using the “betareg” package of R [19].

## 3. Results

### 3.1. Description of Studies

We included 42 studies in our TRIPOD rating. Forty studies [20,21,22,23,24,25,26,27,28,29,30,31,32,33,34,35,36,37,38,39,40,41,42,43,44,45,46,47,48,49,50,51,52,53,54,55,56,57,58,59] were adopted from the systematic review about risk prediction models for melanoma published in 2020, whereas the remaining two studies [60,61] arose from the updated literature search. Details on the information extracted from the studies are summarized in Table A1. Out of the 42 studies, 35 (83%) reported the development of a melanoma risk prediction model, while seven studies (17%) described both the development and external validation of a risk prediction model. None of the studies belonged to the third study type covered by TRIPOD, namely those studies that describe exclusively the external validation of an already published model. The studies were published between 1988 and 2021, with a marked increase in the number of studies in the last decade of this interval. Study designs used were mainly case–control (*n* = 30) and cohort (*n* = 10). Two studies used published material from meta-analyses to determine their risk estimates. The median journal impact factor was 12.5 (range 0–26). The journals in which the studies were published belong to the following subject categories: “Oncology “(N = 15), “Dermatology” (N = 12), “Medicine, General & Internal” (N = 4), “Multidisciplinary Sciences” (N = 4), “Public, Environmental & Occupational Health” (N = 2) and “Other” (N = 3). The category “Other” includes one study [57] that was published in a conference proceeding, one study [41] published in a journal that was not included in the Web of Science Core Collection and one study [53] published in a journal of the category “Biochemistry & Molecular Biology”.

### 3.2. Reporting Completeness per Study in TRIPOD

Figure 1 shows the empirical distribution function of the TRIPOD adherence score. The median adherence to TRIPOD was 57% with a range from 29% to 78%. There was no study that satisfied all requirements of transparent reporting. In total, 34 studies (81%) fulfilled at least 50% of the TRIPOD components, whereas only 3 studies (7%) reached an adherence of 75% or more. A more complete reporting was seen for studies with a combined reporting of model development and external validation (median: 64%; range: 38–78%) compared to development studies, which had a median adherence of 56% (29–75%). However, the score difference was not statistically significant (*p* = 0.11). Studies that claim to report according to the TRIPOD statement (N = 3) [44,47,55] achieved a significantly higher adherence than studies that did not (75% vs. 56%, *p* < 0.05). The lowest adherence to TRIPOD (29% and 30%) was observed in two studies [40,58] which were not published as regular original articles.

### 3.3. Reporting of Individual TRIPOD Components

Completeness of reporting of individual TRIPOD components over all 42 studies and per type of study is shown in Figure 2. In total, 15 of the 37 sub-/items were fulfilled in less than half of the studies for which they were applicable. Two subitems (5c and 10e) were rated as “not applicable” in all studies for which they were rated (item 10e has to be rated only for studies reporting development and validation). Consequently, no adherence score could be determined for these subitems. No sub-/item was reported in all studies, but several sub-/items (3a, 3b, 4a, 5a, 7a and 18) were provided for all seven development and validation studies.

The most and least frequently reported sub-/items are shown in Table 2. Six sub-/items were reported in 90% or more of the studies in which they were applicable. The most frequently reported sub-/items with a relative frequency of 98% were related to model objectives (subitem 3b), study design and source of data (subitem 4a), and sample size (item 8). The least reported sub-/items with relative frequencies less than 10% were those related to blinding methods for predictor assessment (subitem 7b), title and abstract (items 1 and 2), model development procedure (subitem 10b) and model updating (item 17). Item 17 was not reported by any study, but it should be noted that it was only applicable in four studies.

### 3.4. Temporal Analysis and Multivariable Regression

To avoid bias, only original articles were retained for the following analyses, as the low adherence of the two excluded studies [40,58] is due to their publication type. The relationship between the adherence to TRIPOD and the publication year of the study is illustrated in Figure 3. A slight increase in the score over the years could be noted. However, the association was not significant (*p* = 0.078) in a simple beta regression model containing publication year as the only explanatory variable. The variance of the TRIPOD score increased strongly among studies that were published after 2010 compared to studies published before. Appendix A shows the relationship between the TRIPOD adherence and the publication year in different subgroups. The temporal relationship is greater when only studies in journal subject categories “Dermatology” and/or “Oncology” are considered in the model.

When adding the impact factor and the journal subject category as additional independent variables, results of multivariable beta regression revealed a significant influence of the publication year on the adherence score (*p* < 0.001) and the variability of the score (*p* < 0.001). In addition, the journal subject category closely missed significance (*p* = 0.065). The categories “Dermatology” and “Oncology” were associated with higher adherence scores than the categories “Multidisciplinary Sciences”, “Public, Environmental & Occupational Health” and the combined category “Other”. This is shown by the model adjusted mean values of the TRIPOD overall adherence scores calculated for the mean pattern of other variables in Table 3 and further illustrated in Figure 4, in which we have added the journal subject categories to the previous correlation plot. The subject category “Dermatology” has a model adjusted mean score of 62%, while in the subject category “Other” it is only 48%. The journal impact factor had a negligible influence on the reporting quality (*p* = 0.72). An illustration of the relationship between the adherence to TRIPOD and the impact factor is given in the Appendix A.

## 4. Discussion

Our results show substantial deficits in the reporting of risk prediction models for cutaneous melanoma. More than half of the components deemed essential for good reporting in publications of prediction models according to the TRIPOD statement were insufficiently reported. Yet, transparent and accurate reporting is essential in order to be able to interpret the results, appraise study validity, replicate the model, and evaluate its applicability.

### 4.1. Interpretation of Results

Of note, none of the 42 studies included in the analysis satisfied all required TRIPOD components. The maximum adherence score, achieved by the study of Vuong et al. [43] was 78%. Elements of the introduction related to the background and objectives of the study (subitems 3a and 3b) were adequately reported by almost all studies (93% and 98%). The description of the source of data (subitems 4a and 4b) was also at an overall high reporting level (98% and 88%), as well as the discussion of limitations and discussion of results (sub-/items 18 and 19b, 95%). At the same time, several sub-/items were missing in a large proportion of the studies. These include the items related to title and abstract (items 1 and 2), which at first glance seem easy to fulfill, but which comprise a long list of precisely defined elements that all have to be met. For item 1, it is required that the title contains (1) the terms “development” or “validation” (or synonyms) depending on the study type, (2) terms like “risk prediction/risk model/risk score” (or synonyms), (3) the target population, and (4) the outcome that is predicted. In fact, only one of the 42 studies [54] fulfilled all four elements. Other components that were sparsely reported, especially in development studies, are sub-/items related to missing data (item 9) and statistical analysis methods (10a–e), as well as details of model specification and model performance (subitems 15a, 15b and 16). Instructions on how to use the model, for example, were found in less than half of the studies (43%). Reporting in these domains needs to be improved, otherwise re-validating and re-calibrating the developed models in future research and their use in clinical settings will be nearly impossible.

Our results indicate that the mean reporting quality has slightly improved over the past 40 years. However, the variance has also increased sharply since 2010. While there are some recent studies with relatively high scores above 70% [43,44,47,49,54], there are also few recent studies that scored well below average (37% and 38%) [52,57]. Although it might be conceivable that studies published in methodologically oriented journals place a higher focus on the complete reporting of their methods and results, this could not be proven. However, we identified only two studies published in journals of the journal subject category “Public, Environmental & Occupational Health” as our surrogate for methodologically oriented journals which limits assessment of this hypothesis. On the positive side, studies citing TRIPOD had a significantly better adherence than the rest of the studies. Of 12 studies that appeared after the publication of the TRIPOD statement and that could have used the guideline for their reporting, only three studies stated compliance with the guideline explicitly. This shows that the use of established reporting guidelines like TRIPOD needs to increase among researchers conducting risk prediction studies. Ultimately, the goal should be for all publishing researchers to adopt existing reporting guidelines as a matter of course. Scientific journals play a key role in achieving this goal. If journals introduced a mandatory requirement to include a completed TRIPOD checklist with each submitted manuscript describing studies developing and/or validating prediction models, this would increase awareness of TRIPOD and have a positive impact on the quality of reporting. The experiences gained in implementing reporting guidelines such as CONSORT [1] and PRISMA [3] can serve as a role model [62,63].

### 4.2. Comparison with Other Studies

Even though this analysis related to a special segment of prediction studies, our finding of inadequate reporting in the field of melanoma risk prediction is comparable to other studies [64,65,66,67]. Heus et al. [65] included 146 publications across 37 clinical domains and reported a median TRIPOD adherence of 44%, which is even lower than the median of the studies included in our analysis (57%). Other publications [64,67] found poor reporting quality of prognostic models in oral health and for COVID-19. Their results demonstrate that incomplete and non-transparent reporting is an interdisciplinary problem and present in most areas of medicine.

One previous study [66] focused already on melanoma prediction studies and claimed to assess TRIPOD adherence in this area. However, due to a less sensitive search strategy combined with broad eligibility criteria this study is not comparable to ours. The set of studies assessed by Yiang et al. [66] comprised a mixed bag of investigations devoted to predicting (i) occurrence of melanoma in population settings, (ii) progression of melanoma in clinical settings, (iii) survival of melanoma patients, (iv) lymph node positivity of melanoma patients, and (v) correct identification of melanoma in diagnostic settings. Due to the broader definition of melanoma prediction, Yiang et al. [66] should have found many more publications for their TRIPOD assessment than we, but in fact they identified only 27 studies in their literature search. Altogether 34 of the 42 studies in our evaluation were not included in their set of studies, while 19 of their studies were intentionally not covered by us as they did not address prediction models for melanoma occurrence. Interestingly, although the scope of the two investigations on TRIPOD adherence in melanoma prediction studies was different, the results were similar: the median TRIPOD adherence scores in both investigations were nearly identical (61% vs. 57%).

Only one study reported a relatively good TRIPOD compliance (median 74%) related to prediction models for hepatocellular carcinoma [68]. Nevertheless, the study also highlighted several sub-/items that were poorly reported like item 2 (abstract), subitem 10d (specification of measures used to assess model performance in methods section) and subitem 13b (description of the characteristics of the participants).

The item-specific frequency of reporting varied among studies. Items that were very poorly reported in our analysis were very well presented in other studies and vice versa. Nevertheless, the basic statement that reporting of studies about prediction models needs to be improved was the same across all clinical domains.

### 4.3. Evaluation of TRIPOD Feasibility

During the rating process, we identified some limitations of the TRIPOD assessment form. The main feasibility problem was that TRIPOD is primarily designed for longitudinal studies, while we identified mostly case–control studies (N = 30, 71%) in our setting which is quite typical for studies developing prediction models for the occurrence of specific cancer entities. Some elements of the assessment form are not applicable to case–control studies in their current form. This includes the question about the time point of outcome assessment (element of subitem 6a). Since the outcome of case–control studies is already known at the time of recruitment, the element should actually be rated as “not applicable”, but the assessment form does not provide this response option for this element. The same applies to the questions about the reporting of a follow-up time (element of subitem 13a) and the number of participants with missing data for the outcome (element of subitem 13b). Again, “not applicable” is missing as a response option. Furthermore, reporting of participant flow (element of subitem 13a) is very uncommon for case–control studies as opposed to cohort studies. It is thus clear that the TRIPOD assessment form is not optimal in its current form for all study designs.

Furthermore, it can be concluded that TRIPOD is less suitable for publications that do not have the typical structure of introduction, methods, results and discussion. The two studies which were not published as regular original articles [40,58] achieved the lowest adherence scores of the 42 included studies (29% and 30%). However, due to their different structure, neither could report on all components required by TRIPOD. Extensions of the TRIPOD statement will improve the rating of different publication and study types. In 2020 the authors of TRIPOD published an additional checklist that applies explicitly to journal and conference abstracts and contains 12 items [69]. In addition, an extension for clustered data, e.g., datasets consisting of multiple centers or countries, will appear in Spring 2022 [70]. Due to the reasons already mentioned in the last paragraph, an extension or modification of TRIPOD for case–control studies would also be useful.

Another aspect is that TRIPOD may overestimate gaps in reporting by weighting all components equally for the calculation of the score. However, reporting or not reporting has different effects depending on the sub-/item. Regarding item 1, omitting certain terms such as “development” or “validation” from the title affects how well the study is found in literature searches. Furthermore, if a study does not report all predictors that they used (element of subitem 7a), it is no longer possible to apply, replicate or validate the model. Not reporting the source of funding and the role of the funders (item 22) reduces the adherence score in the same way as the two examples before, but the impact is completely different. In addition, some TRIPOD components consist of several individual elements, whereas other items comprise only one or two elements. They, therefore, take different amounts of time and effort to fulfill. For example, item 2 (abstract) includes 12 elements. If a single element, e.g., the specification of a calibration measure, is not fulfilled, although all other eleven elements are present in the abstract, the whole item is considered as not fulfilled. In contrast, only one element (description of study design/data source) is needed to obtain the adherence point of subitem 4b. In addition, it is not taken into account whether the missing element could be reported at all. When a study does not evaluate the calibration of its model, evidently a calibration measure cannot be reported in the abstract. However, according to the assessment form, the element cannot be rated “not applicable”. Thereby, a lack of calibration is penalized twice, in item 2 (abstract) and item 16 (performance measures).

While TRIPOD claims to not provide a measure for the quality of the studies [7], there are nevertheless some items that are related to methodological quality, as already seen in the example of the calibration measure. Another example is whether internal validation was reported (element of item 10b). If this element is not reported, regardless of whether internal validation has actually been performed and could thus be reported or not, the element must be rated “no”. To be more independent of the methodological quality of the studies, it is necessary that especially items related to the analysis and the results can be rated “not applicable”. In fact, it is impossible to completely separate reporting quality from methodological quality, as this would require consistent conditions. Specifically, the studies would all need to have adopted the same methodologic concept and the same statistical analyses, only then TRIPOD would evaluate which study reports more carefully and more completely. Therefore, although TRIPOD is intended to reflect only reporting quality, it is also an implicit indicator of study quality.

If journals required a TRIPOD checklist for all studies describing development and/or validation of prediction models, indicating which items were fulfilled, not applicable or not relevant to the study type, and published the checklist together with the study report after review, it would be easier for other researchers or users of the prediction model to evaluate and interpret the results of the study.

### 4.4. Limitations

The set of studies assessed in our rating comprised the scientific literature on risk prediction models for melanoma identified by three systematic reviews on the topic and our literature update. However, due to the eligibility criteria of the systematic reviews only studies describing solely development and studies describing both development and external validation of risk prediction models for melanoma were included in our assessment. Thus, publications describing exclusively external validation studies of preexisting models were not part of our investigation. In consequence, our results do not allow conclusions about reporting the quality of external validation studies of risk prediction models for melanoma.

Although we felt that defining the overall TRIPOD adherence score as a simple sum score with equally weighted components had its shortcomings and did not adequately reflect reporting quality, we refrained from using a modified score version in which more important components receive a higher weight than less important components. This decision was made to allow for better comparability of our assessment results with other TRIPOD assessments.

Some parts of the TRIPOD assessment form do not lend themselves to a clear objective rating, they contain a subjective flavor. Different raters will thus come to different conclusions on how to rate the corresponding TRIPOD sub-/item. Therefore, it cannot be ruled out that another group of raters would have come to other results regarding the distribution of TRIPOD adherence scores in our set of studies. We have tried to minimize this rater dependence by holding consensus meetings to resolve discrepant ratings and by involving two independent referees in case of persisting disagreement.

## 5. Conclusions

In conclusion, the current level of reporting of risk prediction models for cutaneous melanoma is insufficient, especially with regard to details of the title, abstract, blinding, model-building procedures and model performance. Even though completeness of reporting has increased slightly over the years, there is still much room for improvement. One point that needs to be addressed in order to improve the reporting quality in future research is the more frequent use of the TRIPOD guideline, which is currently rather rare. Otherwise, potentially valuable risk prediction models may be less useful in clinical practice simply because of inadequacies in their reporting.

## Figures and Tables

**Figure 1 healthcare-10-00238-f001:**
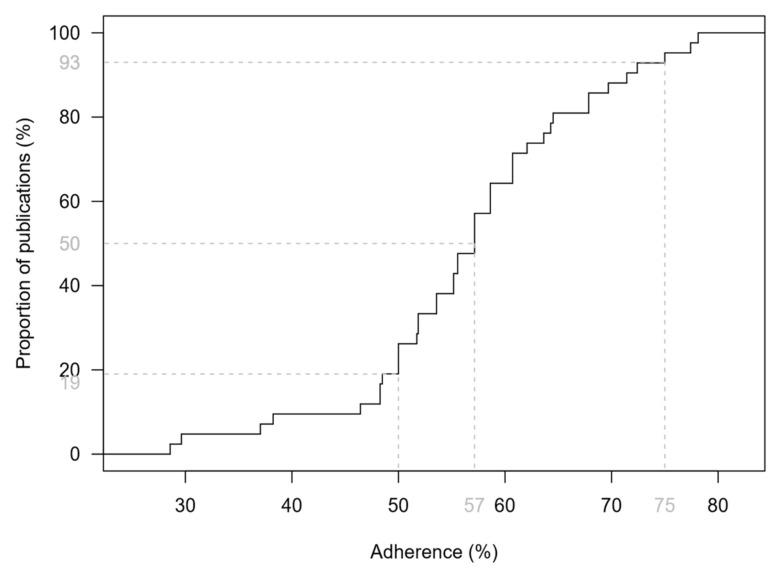
Empirical distribution function of the TRIPOD adherence score based on 42 studies addressing melanoma risk prediction models and their validation. Dashed lines indicate the median, as well as the proportions of studies that achieved a score of less than 50% and less than 75%.

**Figure 2 healthcare-10-00238-f002:**
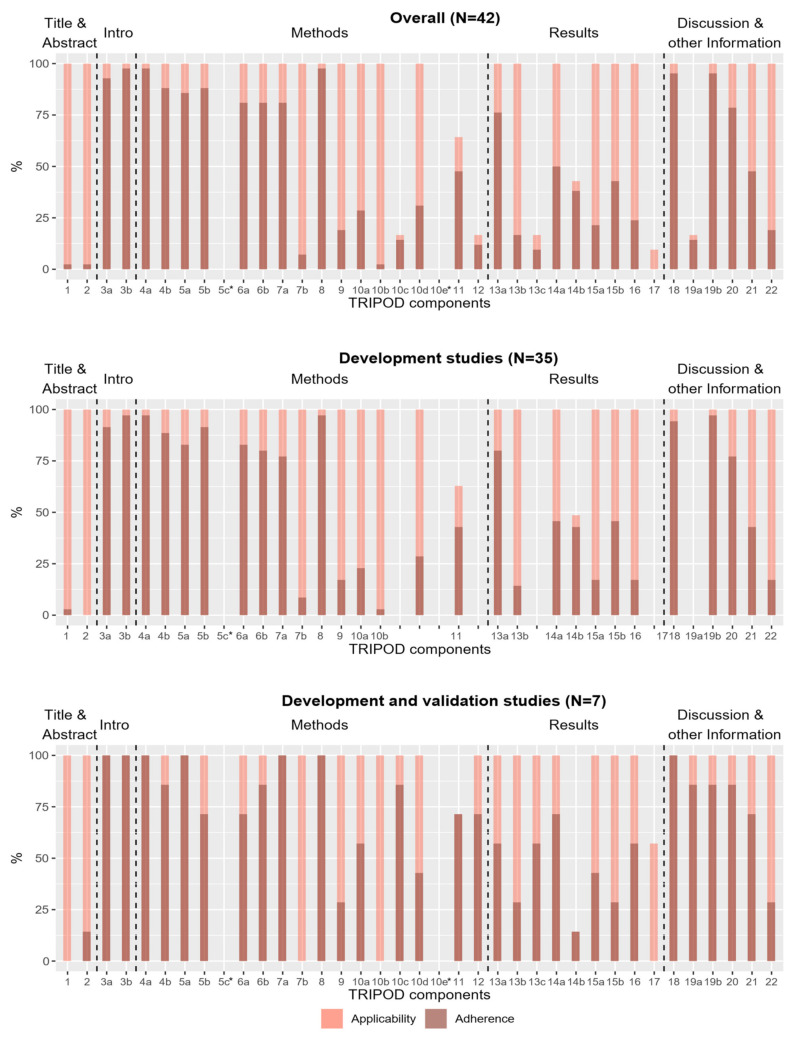
Applicability and reporting of TRIPOD components in the total group of studies (N = 42), in development studies (N = 35) and in development and validation studies (N = 7). Bright bars represent the percentage of studies for which the components were applicable. Dark bars represent the percentage of studies in which the TRIPOD component is fulfilled. * The subitems were rated as “not applicable” in all studies. (Subitem 10e does not apply to development studies, so in this case “all studies” means all development and external validation studies (N = 7)).

**Figure 3 healthcare-10-00238-f003:**
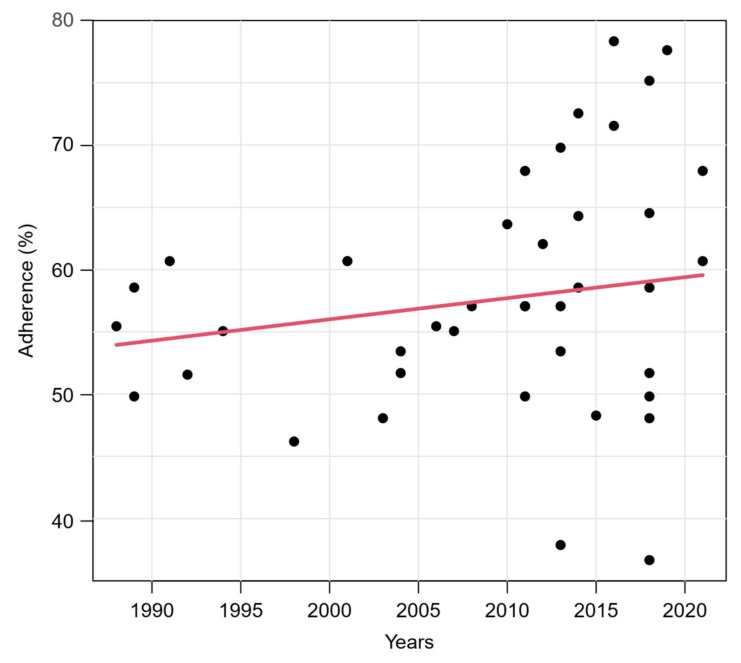
Relationship between TRIPOD adherence and publication year of studies. Red line represents predicted mean curve from a beta regression model based on 40 studies (two studies were excluded, see text).

**Figure 4 healthcare-10-00238-f004:**
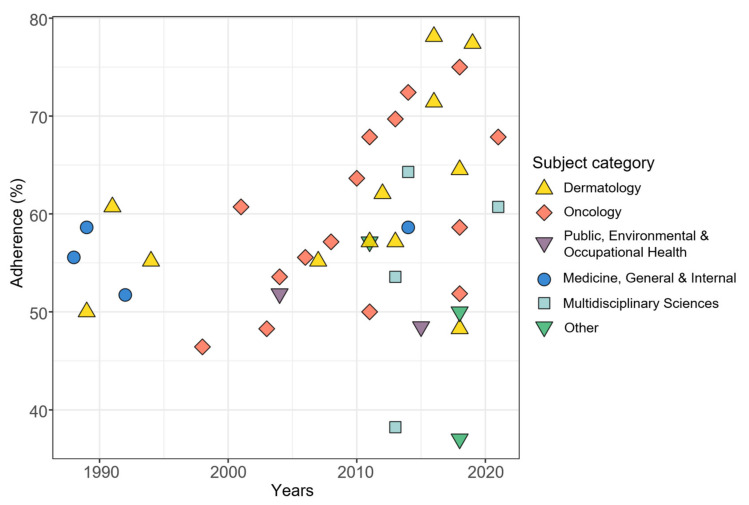
Relationship between TRIPOD adherence and publication year of studies with journal subject categories added. N = 40 (two studies were excluded, see text).

**Table 1 healthcare-10-00238-t001:** Components of the Transparent Reporting of a multivariable prediction model for Individual Prognosis or Diagnosis (TRIPOD) statement adapted from www.tripod-statement.org/ (accessed on 21 December 2021). Items are numbered and subitems are marked with letters.

Title and abstract	
1. Title (D, V)	Identify the study as developing and/or validating a multivariable prediction model, the target population, and the outcome to be predicted.
2. Abstract (D, V)	Provide a summary of objectives, study design, setting, participants, sample size, predictors, outcome, statistical analysis, results, and conclusions.
**Introduction**	
3. Background and objectives	
a. (D, V)	Explain the medical context (including whether diagnostic or prognostic) and rationale for developing or validating the multivariable prediction model, including references to existing models.
b. (D, V)	Specify the objectives, including whether the study describes the development or validation of the model or both.
**Methods**	
4. Source of data	
a. (D, V)	Describe the study design or source of data (e.g., randomized trial, cohort, or registry data), separately for the development and validation data sets, if applicable.
b. (D, V)	Specify the key study dates, including start of accrual; end of accrual; and, if applicable, end of follow-up.
5. Participants	
a. (D, V)	Specify key elements of the study setting (e.g., primary care, secondary care, general population) including number and location of centres.
b. (D, V)	Describe eligibility criteria for participants.
c. (D, V)	Give details of treatments received, if relevant.
6. Outcome	
a. (D, V)	Clearly define the outcome that is predicted by the prediction model, including how and when assessed.
b. (D, V)	Report any actions to blind assessment of the outcome to be predicted.
7. Predictors	
a. (D, V)	Clearly define all predictors used in developing or validating the multivariable prediction model, including how and when they were measured.
b. (D, V)	Report any actions to blind assessment of predictors for the outcome and other predictors.
8. Sample size (D, V)	Explain how the study size was arrived at.
9. Missing data (D, V)	Describe how missing data were handled (e.g., complete-case analysis, single imputation, multiple imputation) with details of any imputation method.
10. Statistical analysis methods	
a. (D)	Describe how predictors were handled in the analysis.
b. (D)	Specify type of model, all model-building procedures (including any predictor selection), and method for internal validation.
c. (V)	For validation, describe how the predictors were calculated.
d. (D, V)	Specify all measures used to assess model performance and, if relevant, to compare multiple models.
e. (V)	Describe any model updating (e.g., recalibration) arising from the validation, if done.
11. Risk groups (D, V)	Provide details on how risk groups were created, if done.
12. Development vs. validation (V)	For validation, identify any differences from the development data in setting, eligibility criteria, outcome and predictors.
**Results**	
13. Participants	
a. (D, V)	Describe the flow of participants through the study, including the number of participants with and without the outcome and, if applicable, a summary of the follow-up time. A diagram may be helpful.
b. (D, V)	Describe the characteristics of the participants (basic demographics, clinical features, available predictors), including the number of participants with missing data for predictors and outcome.
c. (V)	For validation, show a comparison with the development data of the distribution of important variables (demographics, predictors and outcome).
14. Model development	
a. (D)	Specify the number of participants and outcome events in each analysis
b. (D)	If done, report the unadjusted association between each candidate predictor and outcome.
15. Model specification	
a. (D)	Present the full prediction model to allow predictions for individuals (e.g., all regression coefficients, and model intercept or baseline survival at a given time point)
b. (D)	Explain how to use the prediction model.
16. Model performance (D, V)	Report performance measures (with confidence intervals) for the prediction model.
17. Model-updating (V)	If done, report the results from any model updating (e.g., model specification, model performance, recalibration)
**Discussion**	
18. Limitations (D, V)	Discuss any limitations of the study (such as nonrepresentative sample, few events per predictor, missing data)
19. Interpretation	
a. (V)	For validation, discuss the results with reference to performance in the development data, and any other validation data.
b. (D, V)	Give an overall interpretation of the results considering objectives, limitations, results from similar studies and other relevant evidence.
20. Implications (D, V)	Discuss the potential clinical use of the model and implications for future research.
**Other Information**	
21. Appendix A (D, V)	Provide information about the availability of supplementary resources, such as study protocol, web calculator, and data sets.
22. Funding (D, V)	Give the source of funding and the role of the funders for the present study.

D, V: Sub-/Item applies to the reporting of model development and model validation; D: Sub-/Item only applies to the reporting of model development; V: Sub-/Item only applies to the reporting of model validation.

**Table 2 healthcare-10-00238-t002:** TRIPOD components reported in more than 90% and less than 10% of the studies. Completeness of reporting of the sub-/items is given as percentage. Additionally, the number of studies that adhered to the specific sub-/item (n) and the number of studies in which the sub-/item is applicable (N) are provided.

Most Frequently Reported Sub-/Items		% (n/N)	Least Reported Sub-/Items		% (n/N)
3b	Specify the objectives, including whether the study describes the development or validation of the model or both	97.6 (41/42)	7b	Report any actions to blind assessment of predictors for the outcome and other predictors	7.1 (3/42)
4a	Describe the study design or source of data (e.g., randomized trial, cohort, or registry data), separately for the development and validation data sets, if applicable	97.6 (41/42)	1	Identify the study as developing and/or validating a multivariable prediction model, the target population and the outcome to be predicted	2.4 (1/42)
8	Explain how the sample size was arrived at	97.6 (41/42)	2	Provide a summary of objectives, study design, setting, participants, sample size, predictors, outcome, statistical analysis, results and conclusions	2.4 (1/42)
18	Discuss any limitations of the study (such as non-representative sample, few events per predictor, missing data)	95.2 (40/42)	10b	Specify type of model, all model-building procedures (including any predictor selection) and method for internal validation	2.4 (1/42)
19b	Give an overall interpretation of the results, considering objectives, limitations, results from similar studies and other relevant evidence	95.2 (40/42)	17	If done, report the results from any model updating (e.g., model specification and model performance)	0.0 (0/4)
3a	Explain the medical context (including whether diagnostic or prognostic) and rationale for developing or validating the multivariable prediction model, including references to existing models	92.9 (39/42)			

**Table 3 healthcare-10-00238-t003:** Model adjusted mean TRIPOD overall adherence scores for each journal subject category using the mean pattern of other variables.

Journal Subject Category	Model Adjusted Mean TRIPOD Overall Adherence Score in %
Dermatology	62.4
Oncology	58.0
Public, Environmental and Occupational Health	52.4
Medicine, General and Internal	60.5
Multidisciplinary Sciences	51.2
Other	48.4

## Data Availability

The data presented in this study are available on reasonable request from the first author (I.K.).

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
