# Peer review of "Reporting Quality of Studies Developing and Validating Melanoma Prediction Models: An Assessment Based on the TRIPOD Statement"

_healthcare, 2022, doi:10.3390/healthcare10020238_

Round 1
Reviewer 1 Report
This is a very well written study entitled "Reporting Quality of Studies Developing and Validating Melanoma prediction Models: An Assessment Based on the TRIPOD Statement".
The authors conducted a through study investigating the TRIPOD quality metrics in 42 previously published studies in the literature. The findings are illustrated in a concise and clear fashion and the authors adequately discuss the results and the limitations of their study. This study also helps in spreading awareness of the TRIPOD metrics for readers.
The suggestion I have for the authors is to use figure 2 to not only show the percentage of each specific component that was applicable. They can also use it to demonstrate the percentage of compliance. My suggestion is to show overlying each bar (with a different color) the percentage of actual compliance of each category, that way the readers can "visualize" the degree of adherence to each element.
Reviewer 2 Report
Research of this type, as presented in this manuscript, is extremely important in the context of the increasing incidence and mortality caused by various diseases, especially cancers. I fully agree with the authors' statement that only reliably described articles, that meet the reporting guidelines, can be useful for meta-analyzes and comparing the results of various studies.
With this being said, I have a few comments and suggestions for improving this manuscript:
- The introduction lacks information as to why the authors analyzed the prediction models data for melanoma. It would be useful to briefly elaborate on why melanoma is better to be detected at an early stage (quick entry into the following stages of progression, frequent development of resistance to monotherapy, etc.)
- The sentence in lines 83-86 is not clear.
- In figure 3 the authors fitted the regression lines for the collected data, a similar operation would be needed in figure 4 especially for the two most abundant data series 'Dermatology' and 'Oncology'. Comparing the regression line will show the real effect on the adherence score if you limit the categories. The authors wrote 'The categories “Dermatology” and “Oncology” were associated with higher adherence scores than the categories "Multidisciplinary Sciences", "Public, Environmental & Occupational Health" and the combined category "Other".' (lines 243-246) - but what exactly was this impact, how much did it increase?
- The authors wrote 'The journal impact factor had a negligible influence on the reporting quality.' (lines 247-248) - I believe that even if this comparison did not show any difference it would be interesting to show it on a graph.
- Referring to the authors' opinion from the 409-413 line, personally, I would be interested to see the score of the modified score version, for example, of a selected group of articles like 'Dermalogy', to compare how the adherence score differs.
- Some small corrections in the text should be made: italics and/or punctuation (e.g.,)(vs.)(i.e.,)
In conclusion, besides my suggestions, the article is quite well-written, interesting results are shown with high importance for the public and I recommend to accept it after minor revision.
Reviewer 3 Report
The authors' aim was to evaluate the reporting quality of studies, developing and validating risk prediction models for melanoma according to the TRIPOD (Transparent Reporting of a multivariate prediction model for Individual Prognosis Or Diagnosis) checklist. They finally assessed 42 studies including 35 studies reporting the development of a prediction model and 7 studies reporting both development and validation.
There is substantial heterogeneity in published risk prediction models for melanoma and direct comparisons between models are very difficult, therefore external validation is largely missing. It is important the adherence to reporting guidelines like TRIPOD. And this, should be obligatory in order to achieve a solution of this problem. So, the subject of the study is relevant. The article is well written and the methodological approach and the way of presenting the data is good
